# The Role of Ultrasonography in Hip Impingement Syndromes: A Narrative Review

**DOI:** 10.3390/diagnostics13152609

**Published:** 2023-08-07

**Authors:** Panagiotis Karampinas, Athanasios Galanis, John Vlamis, Michail Vavourakis, Eftychios Papagrigorakis, Evangelos Sakellariou, Dimitrios Zachariou, Spyridon Karampitianis, Elias Vasiliadis, Spiros Pneumaticos

**Affiliations:** 3rd Department of Orthopaedic Surgery, National & Kapodistrian University of Athens, KAT General Hospital, 14561 Athens, Greece; karapana@yahoo.com (P.K.); athanasiosgalanis@yahoo.com (A.G.);

**Keywords:** hip, femoroacetabular impingement, ultrasound, ultrasonography, groin pain, acetabulum, femoral head, hip pain, review

## Abstract

Hip pain is indubitably a frequent clinical problem deriving from copious etiologies. Hip impingement syndromes are one of the most prevalent causes of persistent groin pain, especially in young and active patients. Diligent imaging of the hip region is indispensable to discern femoroacetabular impingement, as the differential diagnosis of hip pain can be exceedingly arduous. Despite hip radiography being plain and broadly attainable, it offers narrow information concerning soft tissue pathologies around the hip joint (extra-articular hip impingement syndromes). Magnetic resonance imaging and arthrography remain the gold standard examination for detecting intra-articular pathologies; however, they are widely considered expensive, time-consuming and characterized by confined. Consequently, ultrasonography has emerged as an alternative valuable diagnostic tool for distinguishing the underlying abnormalities that trigger femoroacetabular impingement. Proper hip ultrasound examination provides dynamic assessment, while also beneficial for guided intervention around the hip joint. Ultrasound hip examination is exacting due to its complex regional anatomy and deep location. It is capable of providing detailed information about various hip quadrants. An adept operator can identify both intra-articular and extra-articular pathologies. In addition, with ultrasonography, hip injections have been rendered relatively undemanding, aiding in therapeutic and diagnostic purposes. This paper aims to provide a succinct and compendious review of the existing literature, accentuating the crucial role of ultrasonography in diagnosing hip impingement syndromes and determining whether an additional examination is required regarding distinguishing between intra-articular and extra-articular syndromes.

## 1. Introduction

Hip pain is unexceptional in all ages and activity levels [1]. Differential diagnosis of hip pain is of paramount importance for determining efficacious therapy. Circumstantial medical history and physical examination can distinguish the causes [2]. Hip pain can be pinpointed in one of three quadrates: anterior, lateral, or posterior, whilst the root of pain may be intra-articular or extra-articular. Predominantly, pain is featured as groin pain (83%) and is reproduced in extreme flexion and rotation of the hip [1,2]. Prolonged sitting in a car, getting in and out of a car or rising from a chair are customary pain-triggering conditions. Hip impingement syndromes are common causes of hip pain in active young adults from 20 to 50 years of age [2,3].

Femoroacetabular impingement (FAI) is a syndrome correlated to sports or traumatic aetiology. There are three acknowledged intra-articular causes of FAI. Firstly, cam deformity is the bony overgrowth of the femoral head and neck. Pincer deformity of the acetabulum (too much coverage of the femoral head) is the second cause of hip impingement, whereas the third situation is the co-occurrence of both deformities [2,3,4,5]. Intra-articular hip pain is localized by the patient using the C sign. Physical examination maneuvres (FABER and FABIR tests) are sensitive to FAI pathology but are not specific. Additionally, there are several extra-articular causes commonly provoking hip impingement pathologies and groin pain, with corresponding clinical examination and tests to intra-articular ones. However, extra-articular disorders may coexist intra-articularly [4,6]. Athletes presumably require surgical intervention for these conditions, notably those enduring both deformities [1,7].

Standing anteroposterior hip and pelvic radiography is the primary imaging test, accompanied by magnetic resonance imaging (MRI) or ultrasonography (US), that may contribute conspicuously to the differential diagnosis in conjunction with past medical history and physical examination findings [1,3]. Standing anteroposterior pelvic radiographs are fruitful for diagnosing pincer deformity. Meyer lateral and 90-degree Dunn views can comprehensibly depict cam deformity [3,5,6]. Plain radiography is a simple and cost-effective imaging technique but cannot provide sufficient information about cartilage and labrum injuries, whilst magnetic resonance arthrography, which is irrefutably regarded the gold standard examination for diagnosing hip impingement syndromes, is considered eminently accurate but also tremendously expensive and with limited availability. High-resolution ultrasonography (US) can be implemented to evaluate hip pathology in adults and, more customarily, in children. Due to its size and deep location, the hip joint examination can be exceedingly strenuous. US has been demonstrated as a functional tool in assessing tendons, ligaments, muscles, nerves, synovial recesses, articular cartilage, bone surfaces and joint capsule. The preponderant goals of US imaging are discerning and localizing pathological processes of these structures, discriminating between intra-articular and extra-articular pathology, executing diagnostic and therapeutic interventional procedures and monitoring therapy efficacy [7,8,9]. To our knowledge, perusing the existing literature, the number of papers arguing about the potentially beneficial role of ultrasonography concerning diagnosis and management of hip impingement syndromes is scanty.

This paper aims to review the literature regarding the utilization of ultrasonography in diagnosing hip impingement syndromes and to underline its role in the differential diagnosis of intra-articular and extra-articular syndromes, in a way to proceed, where necessary, to apposite further investigation with MRI or MR hip arthrography and appropriate treatment, including hip arthroscopy.

## 2. Materials and Methods

Literature search was conducted utilizing the MEDLINE/PubMed, Google Scholar, Web of Science and Embase databases for articles published from 1989 up to 2023. Keyword search terms were: “FAI”, “hip impingement syndromes”, “hip impingement”, “hip pain”, “ultrasonography”, “hip imaging” and “hip ultrasound”. Language filters were activated for English. No restrictions were implemented in terms of the scientific articles’ publication date. Inclusion criteria were clinical studies, case series, reviews and papers reporting data pertinent to our topic. Exclusion criteria were papers presenting trials or cases about hip imaging and ultrasonography for conditions different from hip impingement syndromes. Articles in full text were scrutinized to retrieve additional relevant studies. The collected data were entered into an Excel spreadsheet. Extracted data involved authors, gender, diagnosis, patients’ age, publication data, intervention and study location.

## 3. Discussion

### 3.1. Hip Pain and Femoroacetabular Syndromes

Hip and groin pain derives from intra-articular and extra-articular disorders, while it is conventional for both types to coexist [3]. In terms of intra-articular conditions, anomalous offset and sphericity of the femoral head–neck junction may prompt cam-type FAI. On the other hand, acetabular overcoverage (from acetabular protrusion or focal retroversion) may engender pincer-type FAI. Also, some patients may experience FAI-associated symptoms related to a muscle injury in the hip and pubic symphyseal region [7]. Groin pain is a very early complaint of patients suffering from FAI. Retardation in diagnosing hip pain pathologies may be linked to prolonged training interruptions, unnecessary medical and surgical treatments and potentially considerable cartilage damage [10]. Criteria for the diagnosis of FAI are the following: persistent hip pain for >3 months, no clinical affirmation of inflammatory arthritis, hip internal rotation ≤ 20° in 90° of hip flexion, lateral center-edge angle > 20° (i.e., absence of dysplasia), alpha angle > 60° on any plain radiographic view or radial MRI/CT reformat and/or lateral center-edge angle > 40° and/or existence of cranial acetabular retroversion corroborated on MRI/CT, intra-articular pathology verified by a diagnostic injection or MRI evidence of labral-chondral, chondral, or labral injury [4].

Extra-articular syndromes that are suspected causes of hip pain are ischiofemoral impingement (IFI), snapping hip (SH), greater trochanteric pelvic (GTPI) and subspine impingement (SI). Differential diagnosis of hip and groin pain is extensive. It includes injury of the acetabular labrum and hip articular hyaline cartilage, musculotendinous injury of the adductor and rectus abdominis, osteitis pubis, stress fracture, osteoarthritis, osteonecrosis, posterior inguinal wall deficiency, hernia, tumor and infection. Potential causes of hip discomfort and fever involve psoas abscess, prostatitis, pelvic inflammatory disease and urinary tract infection [11]. Differential considerations in imaging regarding groin pain include the comparatively frequent partial or complete muscle tears, with rectus femoris being distinctly vulnerable as it crosses two joints. In particular, kicking sports are recurrently implicated, and the reflected (indirect) head is more commonly injured than the direct head [12].

### 3.2. FAI and Hip Imaging

#### 3.2.1. Hip Imaging in General

Imaging of the hip can be accomplished with radiography, ultrasound, CT, and MRI, as well as CT and MRI after intra-articular contrast administration. Diagnostic injections with an anaesthetic agent yield meaningful diagnostic feedback concerning intra-articular and extra-articular conditions. Imaging and imaging-guided anaesthetic injections are significant for potential surgical planning when considering the convolutedness of hip pain etiology [7].

In the mid-1990s, MRI arthrography was the fundamental examination carried out for evaluating labral tears and remains the imaging gold standard for FAI until nowadays [9]. Magnetic resonance arthrography (MRA) is considered pricey, time-consuming, and characterized by confined availability, while it can potentially generate adverse events [13]. MRA offers dependable information for the evaluation of labral injuries; however, its role in assessing cartilage lesions is comparatively slender. In the past, magnetic resonance arthrography (1.5 tesla) with gadolinium injection was the diagnostic norm regarding detecting labral tears. Notwithstanding, with the contemporary advancements of 3-tesla MRI and specialized hip protocols, a noncontrast 3-tesla MRI is as sensitive and specific as magnetic resonance arthrography [14,15].

#### 3.2.2. Ultrasonography-Associated Benefits and FAI

US features unequivocally some material advantages: lack of radiation, adequate visualization of the joint cavity, quantification of soft tissue abnormalities, capability for multiple joint scannings, noninvasiveness, speed of performance, swift side-to-side anatomic contrast and superior typification of fluid. Furthermore, it is a relatively low-cost imaging tool, patient compliance is augmented, and it can provide a dynamic real-time study of multiple planes [16,17]. Ultrasound can depict even minimal articular cartilage abnormalities, bony cortex, and synovial tissue abnormalities. It is regarded as being more sensitive than X-rays for tracking small osteophytes [16]. Nonetheless, ultrasound examination credibility is determined by the observer’s experience. A versed examiner is requisite for successfully employing the technique and accurate interpretation of the US images, as manifested in other sonographic diagnostic fields [18]. Ultrasonography is a dynamic imaging modality commonly implemented to evaluate extra-articular pathology. Direct patient contact during imaging offers the ability for maneuvers that provoke symptoms to be assessed when performing the test and provides the possibility of executing guided procedures in the hip joint and periarticular soft tissues [3,19,20]. US is a consummate technique for guiding interventional musculoskeletal procedures [21]. US-guided hip injections are advantageous for diagnosing intra-articular and extra-articular syndromes, being safe and portable with no severe complications, while lacking ionizing radiation exposure [3,8]. Pateder and Hungerford noticed hip injections to be 100% sensitive and 81% specific for differentiating hip pathology from lumbar spine pathology [22]. Patients with chondral damage experienced greater relief from injection than those without, irrespective of severity. The occurrence and severity of FAI and labral pathology did not affect the extent of relief from the injection. Concurrent extra-articular pathology did not alter the interpretation of the percent relief from injection. Therefore, the understanding and diagnostic merit of an anaesthetic injection in subjects with primary intra-articular pathology does not require modifications by coexisting extra-articular hip pathology [23].

### 3.3. Ultrasound and Hip Quadrants

#### 3.3.1. Anterior Quadrant

Preponderantly, there are four hip quadrants for evaluation at the hip region. During the US examination of the anterior hip region, four osseous structures are recognized: the anteroinferior iliac spine, acetabular rim, femoral head and femoral neck. The anterior superior labrum can be visualized sonographically as a triangular, echo-bright formation stretching inferiorly from the acetabulum and draping over the femoral head and is the sole structure sufficiently depicted with the US. In addition, the joint capsule and the anterior synovial joint recess can be detected. Effusions, which are most fruitfully estimated along the femoral neck, are spotted as a hypoechoic to anechoic fluid collection that bloats the capsule [8,24]. Joint effusion is discovered when the distance between the anterior layer of the synovium and the femoral neck is larger than 7 mm, or the difference between both hips is larger than 1 mm [24]. In terms of extra-articular structures, the following can be distinguished: the anterior regional muscles (sartorius muscle, tensor fascia lata muscle, rectus femoris iliopsoas and pectineus muscles), the iliopsoas tendon, and the bursa. It is attainable for pinpointing iliopsoas tendinopathy, bursitis and snapping [8,24]. Regarding tendinopathy, the iliopsoas tendon is revealed as hypoechoic, bulgy, and lacking a fibrillary pattern. The iliopsoas bursa is disintegrated and cannot be portrayed in the normal subjects. Owing to 15% of the iliopsoas bursa communicating with the hip joint, iliopsoas bursitis is occasionally linked to hip joint pathology. Atypical iliopsoas tendon motion is the usual ground of extra-articular snapping hip and can be diagnosed by dynamic US examination [24].

#### 3.3.2. Medial Quadrant

The medial hip US scrutiny can discern the femoral neurovascular bundle comprising the femoral vein, artery, and nerve, from medial to lateral, respectively. Moreover, other structures that can be identified are the muscles longus, brevis, and magnus/adductor compartment, along with their myotendinous insertions up to the pubis [8]. Concerning adductor tendinopathy, hypoechoic features and tendon thickening can be noticed when contrasted with the asymptomatic side [24].

#### 3.3.3. Lateral Quadrant

Regarding the lateral examination, the greater trochanter, gluteus minimus and medius tendons and their insertions are observed. The bursae surrounding the greater trochanter is not visible in the US in standard conditions. Yet, it is achievable to detect the fascia lata with the US [8]. The trochanteric bursa is located superficially to the posterior insertion of the gluteus medius tendon and the lateral aspect of the greater trochanter and deep to the gluteus maximus. The foremost reason for greater trochanteric pain syndrome is conventionally thought to be correlated to bursopathy. However, studies employing US or MRI all denoted that hip abductor tendinopathy or tear, chiefly in the gluteus medius tendon, is the primary cause of this syndrome. Extra-articular lateral snapping hip is rooted in intermittent impingement of the posterior border of the fascia lata or the anterior section of the gluteus maximus over the bony protrusion of the greater trochanter. This condition resembles the snapping iliopsoas tendon, but the snapping sensation is perceived laterally [24]. Dynamic imaging with external rotation followed by extension might display a snapping gluteus maximus or iliotibial band over the greater trochanter [16].

#### 3.3.4. Posterior Quadrant

The posterior hip quadrant is scarcely evaluated with the US, being less frequently affected by pathological alterations compared to the other quadrants. The ischial tuberosity is effortlessly evident on the US screen. Structures that can be detected are the conjoined insertion of hamstrings/extensor/ischiocrural tendons, consisting of semimembranosus, semitendinosus, and biceps femoris. On the other hand, the ischiogluteal bursa is imperceptible under US imaging. The sciatic nerve is consistently located on the lateral side of the ischiocrural tendons, posterior to the gluteus maximus [8]. In chronic tendinopathy, the proximal attachment of the tendon becomes apparent as hypoechoic and swollen. Calcification can sporadically be identified at the tendon insertion or within the tendon. Apophyseal avulsion, ordinarily noticed in juvenescence, is visible as a hyperechoic bony fragment and cortex abnormality at the insertion area with adjacent hematoma. Additionally, ischiogluteal bursitis can be discerned, a condition known as “weaver’s bottom” and triggered by prolonged sitting. It also transpires in patients with cachexia or subjects with acute weight loss [25].

### 3.4. Ultrasonography for Intra-Articular Pathology

#### 3.4.1. Labral Tears

Broadly, the most prevalent finding in FAI is labral participation. Hip pain in pincer-type FAI emanates from acetabular labral tears or the interaction between cam morphology and cartilage defects [26]. Labral tears can be described along with FAI, with prevalence estimated at 10 to 15% [12]. The prevalence of cam deformity is 41% in nonprofessional male soccer players and 17% in male nonathletes [27]. Patients with labral tears are typically presented with anterior hip pain and a history of sports-related or traumatic injury. Activities such as dancing, gymnastics, hockey, basketball and soccer are correlated [12]. Most symptomatic patients with characteristics of FAI demonstrated deformities on US examinations, even in the absence of osteoarthritis. Focusing on the pathologic characteristics of FAI, US can amply portray the articular labrum in its anterosuperior aspect, which is mainly the area that is more susceptible and where commencing damage is usually detected [13,28]. Labral tears are most regularly located in the anterosuperior quadrant between the 12:00 o’clock (superior) and 3:00 o’clock (anterior) positions (Figure 1) [7]. The role of ultrasonography in identifying labral pathology is narrow, given the imperfect evaluation of the entire labrum, while it features lower accuracy and sensitivity (44%) contrasted to MR arthrography [7].

US provides small-scale information on cases of bone fractures and labral tears [8]. The contribution of ultrasound in diagnosing labral pathology is exiguous, exhibiting low accuracy and sensitivity when compared with MR arthrography [29]. The technique is still evolving, and more experience is required regarding the interpretation of US examinations [29]. A role for US in diagnosing labral tears derives from a study that concluded that when the impingement test is negative and if a labral tear is still presumed, ultrasound can creditably diagnose most tears of the acetabular labrum. MR arthrography is propounded in cases where the ultrasound is negative, but patients continue to suffer from specific symptoms [28]. During a US examination, the anterosuperior aspect of the acetabulum and labrum (where most FAI lesions are pinpointed) can be adequately visualized [13]. Jin et al. observed that US demonstrated a substantial 82% sensitivity rate for diagnosing labral tears, whereas specificity and accuracy were 60% and 75%, compared with 91%, 80%, and 88%, respectively, for MRA [30]. Labral calcification and a modified echo structure are the most repeated findings. These findings could support FAI evaluation, especially when MRA is not promptly accessible [13].

#### 3.4.2. Cam Impingement

Early and punctual diagnosis of cam-type impingement before the cartilage is damaged is of paramount importance in orthopaedics. Lerch et al. compared the diagnostic value between ultrasound and plain radiographs and deduced that ultrasound is as dependable as plain radiographs in diagnosing cam-type FAI and can serve as an alternative or supplementary method in initial imaging [18]. The anterosuperior osseous contour is more beneficial for assessing cam FAI than the anterior contour. This deduction is aligned with an article by Pfirrmann et al. assessing the distinct location of cam deformities at the femoral head–neck junction [25]. The authors inferred a preponderance of the cam deformity at the anterosuperior aspect compared to the anterior aspect of the femoral neck. Enlarged alpha angles have been recognized as risk factors for hip pain owing to cam impingement. Ultrasound has been freshly utilized to compute alpha angles in diagnosing cam morphology, but its value still needs to be studied. Top-level female adolescent ballet dancers with alpha angles greater than 60° presented worse hip pain and function [26]. A study challenging the association between MRI and ultrasound measurements in patients with cam-type FAI drew the inference that ultrasound may be a functional instrument for the early diagnosis of cam-type FAI in everyday practice. They discovered robust correlations between parameters analyzed (alpha angle, evaluation of head–neck junction region) on MRI and ultrasound [18]. According to Buck et al., cam morphology (alpha angle measurement) can be quantified on longitudinal ultrasound images of the anterolateral aspect of the femoral head–neck junction by applying a five-step course of action [10]. Lerch et al. concluded a strong correlation coefficient (r = 0.76, *p* < 0.0001) in alpha angle figures between MRI and ultrasound [18]. In contrast, Buck et al. decided alpha angle measurements were not advantageous in reaching a clinical diagnosis, especially for cam morphology. The discernment of an anterosuperior cam deformity is sensitive, while the presence of an anterosuperior bony prominence is specific for cam FAI (Figure 2) [10]. Depiction of the femoral neck anterior osseous contour was feasible by utilizing US to detect the structural changes representative of cam-type FAI. The anterior femoral distance (AFD), which is determined by the perpendicular distance between, a line drawn along the cortex of the anterior aspect of the greater trochanter/anterior femoral neck and the point of maximal femoral head–neck overgrowth, might be employed as an additional beneficial feature on US examination for cam-type FAI. This quantitative measurement for cam-type FAI is helpful in descrying cam lesions [10,31]. Lohan et al. revealed that the depth of epiphyseal overgrowth, which is defined as AFD in MR arthrography, is of prime significance in ascertaining the presence of cam-type FAI, while being possibly more suitable than an indirect measure of the alpha angle [32]. The AFD US measurement, notably for the anterior contour of the femoral head–neck junction, offers precious information, thus it could be rendered a useful method for identifying cam-type FAI [31]. Lohan et al. demonstrated a remarkable variability of alpha angle measurements carried out on MR images. Statistically, they inferred no value for alpha angle measurements in determining the presence or absence of cam FAI [32]. Nouh et al. assessed the value of a subjective assessment of the alpha angle on MR images. Measurement of the alpha angle worked as the standard of reference. Owing to the occurrence of the relatively low areas under the curve (≤0.606) of the ROC analysis, they deduced that subjective assessment of alpha angles is not optimal unless one is pretty assured about a bony malformation [33].

### 3.5. Extra-Articular Pathologies and Ultrasonography

Extra-articular entities involving iliopsoas tendonitis, snapping hip, iliotibial band and greater trochanteric bursitis, along with gluteal tendon injury, can also produce atypical symptoms [34]. Anaesthetic and corticosteroid injections can be meticulously executed by implementing ultrasound guidance for dealing with extra-articular pathology [7]. Kivlan et al. recorded roughly 90% pain relief following injection in patients with FAI and extra-articular pathology [23]. Ultrasonographic guidance prevents the patient from radiation exposure and does not demand contrast [35]. Extra-articular US pathology comprises multiple tendon pathology and snapping of the Iliopsoas tendon. Diagnostic ultrasound assesses for iliopsoas snapping, iliopsoas tendon irregularities and iliopsoas bursal distention [7]. Dynamic US scanning can indicate a snapping iliopsoas tendon. Even if the snapping tendon is not discernible sonographically, the patient can benefit from accurate injection into the iliopsoas bursa and detect the pathology. The tendinopathy of the adductor longus tendon, reported as pubalgia, is conspicuous on the US when the tendon is thickened, whilst if it is entirely ruptured, the insertions appear separated from the symphysis pubis. Likewise, tendinopathy, tears/partial and complete/and avulsion of the hamstrings can be perceived. During snapping of the iliotibial band, known as the extra-articular snapping hip, the US can distinguish with maximal accuracy the ground for the clicking as being either the iliotibial band over the greater trochanter or the gluteus maximus muscle, while the fascia lata may look thickened. Dynamic sonography indicates abrupt displacement of the iliotibial band or the gluteus maximus muscle overlying the greater trochanter as a sore snap during hip motion, predominantly during flexion of the adducted extended hip. Bursal pathologies such as iliopsoas and ischiogluteal bursitis can be identified and differentiated [4,24].

Because many intra-articular abnormalities, like labral tears and chondral lesions, are not approachable to US, it exhibits constraints in evaluating patients with groin pain. Ultrasound and ultrasound-guided hip injection demonstrate high accuracy in diagnosing hip impingement syndromes with atypical symptoms [34]. Owing to up to 76% of FAI cases being of a mixed type (cam FAI and pincer FAI combination), and because it is not attainable to address the pincer component with US, additional imaging may be required for a careful evaluation of FAI [10].

### 3.6. Ultrasound-Guided Injections

A diagnostic image-guided intra-articular hip anaesthetic injection can be a good adjunct aiding in attesting that the patient’s pain is associated with an intra-articular hip disorder. Intra-articular hip injections are executed with fluoroscopic or ultrasonographic guidance in an outpatient location [3,4]. Various injection treatments that can be offered by vigilant ultrasonography techniques to patients with hip pain include cortisone injections, hyaluronic acid injections, platelet-rich plasma injections and combined injections. Furthermore, it is cardinal to emphasize that diagnostic procedures such as taking biopsy tissues or hip aspirations for detecting postoperative prosthetic joint infections can be facilitated by ultrasonography. Patients suffering from FAI and mild chondral pathology or acetabular delamination demonstrate extensive pain relief after proper intra-articular anaesthetic injection. This type of response to anaesthetic injection would not be observed with extra-articular pathology. Byrd and Jones displayed 90% accuracy of a positive response to intra-articular injection, corresponding with findings on hip arthroscopy in young adults with pre-arthritic hip disease [36]. Nevertheless, with ultrasonography, verification of intra-articular placement is restricted to visualization of the needle tip and not the entire joint capsule [4]. Ultrasound and ultrasound-guided hip injection present high precision in diagnosing FAI and recognizing intra-articular pathology in patients exhibiting nontypical pain symptoms. US diagnosis’s sensitivity and accuracy of cam impingement were relevantly high; no patient demonstrated true-negative cam impingement nor revealed a true-negative and false-negative labral tear [34]. Levy et al. delineated that the patients with uncommon FAI symptoms manifest comparable notable improvements after hip arthroscopy in terms of outcome scores, postoperative pain and satisfaction compared with patients who demonstrate classic anterior groin pain. Consequently, it is crucial to differentiate between intra-articular and extra-articular pathology [37]. In a systematic review, Khan et al. documented that pain relief acquired from an intra-articular hip injection bolsters a diagnosis of FAI [38].

Contrariwise, it is important to discuss the features of ultrasound-guided interventions contrasted to fluoroscopy-guided injections. In a 2014 study by Byrd JW et al., 50 patients underwent an ultrasound-guided intra-articular hip injection for hip pain deriving from copious etiologies (including FAI), and had previously sustained fluoroscopy-guided intra-articular hip injections by musculoskeletal radiologists. Patients rated the corresponding techniques in terms of convenience and pain. Results overwhelmingly indicated that the vast majority of patients considered ultrasound-guided injections to be more convenient and significantly less painful, connoting that ultrasound-guided intra-articular hip injections can be an efficacious alternative to conventional fluoroscopy-guided techniques [39].

## 4. Conclusions

Ultrasound, dynamic examination and ultrasound-guided aspiration or injection techniques can be featured as noteworthy tools when evaluating and differentiating intra-articular and extra-articular hip impingement pathologies. It is vitally important to be conversant with the intricate anatomy of the hip for diagnostic and therapeutic purposes. It is attainable to primarily assess hip impingement syndromes by plain radiographs and the employment of scrupulous ultrasonographic imaging, which are generally cost-effective methods. Only when more elaborate information is required for affirming the diagnosis of an impingement syndrome, MR imaging should be performed for portraying comprehensive damage of the cartilage and labrum.

## Figures and Tables

**Figure 1 diagnostics-13-02609-f001:**
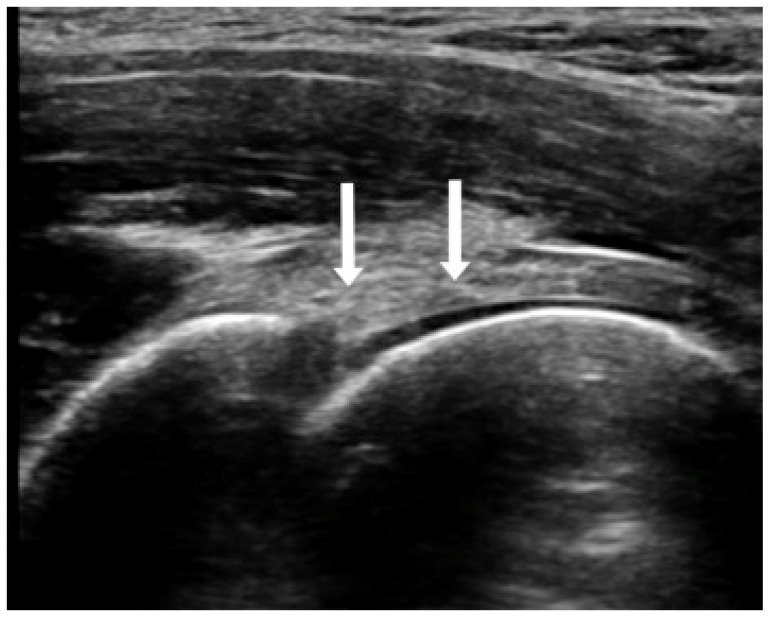
Anterosuperior labral tear indicated by arrows.

**Figure 2 diagnostics-13-02609-f002:**
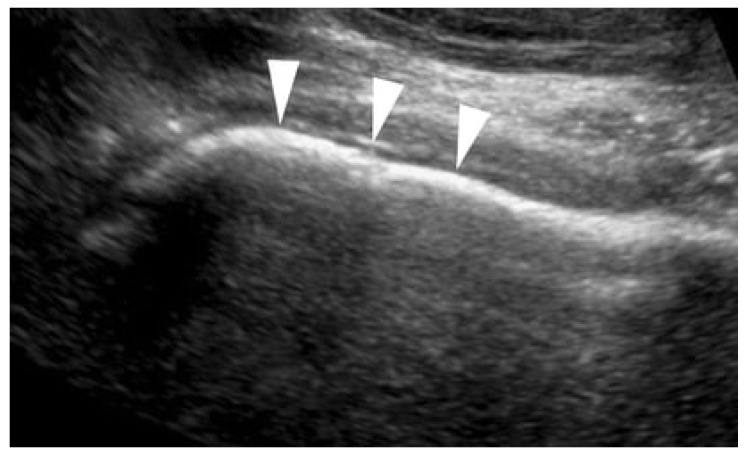
Cam impingement indicated by arrows.

## Data Availability

Not applicable.

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
