# Peer review of "The Role of Ultrasonography in Hip Impingement Syndromes: A Narrative Review"

_diagnostics, 2023, doi:10.3390/diagnostics13152609_

Round 1

Reviewer 1 Report

Thanks for the job. It is a topic that is of great interest to me and I have read it with great pleasure.

However, the main problem is that the work lacks a scientific methodology. It is simply a descriptive review, but its structure does not reflect this type of work either. It must be remembered that narrative reviews represent one of the lowest steps in the pyramid of scientific evidence, and that in any case, they should provide very exhaustive information, being carried out by the greatest experts in the field.

Here are some tips to start improving the organization of the text. However, my main recommendation is that this narrative review provide more comprehensive data on the diagnostic accuracy of ultrasound compared to reference tests (validity) and its reliability...

ABSTRACT

Suggestion: follow a more typical Abstract structure, placing the objective of the study (lines 22-24) previously.

1. INTRODUCTION

- General structure of ideas:

o Paragraph 1 (lines 29-38): talks about hip pain in general, causes, symptoms and diagnosis. This paragraph ends by talking about diagnostic techniques, already presenting the possible usefulness of ultrasound... however, this idea is cut off here, and is taken up again in paragraph 3.

o Paragraph 2 (lines 39-52): talk about FAI (femoroacetabiular impingement), as the most frequent cause of hip pain in young adults. And also from extra-articular causes.

o Paragraph 3 (lines 53-60): talks about ultrasound, presenting its difficulties in the hip and its usefulness. It would be good to end this paragraph with the shortcomings that the bibliography has in this regard, and that this article comes to cover.

o Paragraph 4. Present your study objective.

- 1.1 “Hip pain and Femoroacetabular syndromes”. Once you have already presented your study objective, you should not continue with the INTRODUCTION, however you continue with it. The information provided at this point on "Hip pain and Femoroacetabular syndromes" does not cover the objectives of his study... it is simply introductory on clinical presentation, epidemiology, diagnostic criteria... on these problems.

- 1.2. “FAI and hip imaging” continues within the INTRODUCTION point, giving information on:

or 1.2.1. General imaging tests. I consider that it is not the objective of the study... and that, therefore, it can be part of the INTRODUCTION, but not of the information that it provides based on the OBJECTIVES of the study.

or 1.2.2. ultrasound. This point does fulfill one of the objectives of the study.

- 1.3. “Ultrasound and hip quadrants”: this point seems relevant to me, but it simply presents the most relevant structures to be examined in each quadrant, as a presentation.

2. Ultrasonography for intra-articular pathology

- Methodologically it is difficult for me to understand why we are now talking about point 2, leaving the INTRODUCTION… In fact, point 1.1 should have ended. It addresses two pathologies "Labral tear" and "Cam impingement" adequately.

3. Ultrasound-guided injections

- Despite being one of the objectives of the study, it is extremely confusing to put it between point 2 and point 4, since ultrasound-guided techniques can be performed for both intra- and extra-articular structures.

4. Extra-articular pathology

- It is again confusing for the reader, that it is simply titled "Extra-articular pathology" but it speaks more of a guide to techniques for extra-articular structures, rather than for the diagnosis of these pathologies, as would be expected, following the model of the point 2. However, starting from line 301, he surprisingly talks about these pathologies.

5. Conclusions: The conclusions use overly subjective terms such as “irrefutable important”. It is expected that data on the reliability and validity of ultrasound in relation to its Gold Standards will be offered at this point.

Author Response

A methodological issue was perceived by the reviewer, probably due to misunderstanding, as we had not stated clearly at which point the introduction ended and then at which point the discussions were initiated. We changed that in a clearly visible way. We made several changes to the introduction according to the reviewer's instructions, all of which are evident in the revised text. Regarding point 2, we addressed all the issues raised by clarifying the headings and subheadings of the manuscript. We moved the injections section after the intraarticular and extraarticular parts as instructed. We changed the title of the section elaborating on extraarticular pathologies. Finally, we made alterations to the conclusion according to the reviewer's comments.

Reviewer 2 Report

Reviewer's Report:

Title: The Role of Ultrasonography in Hip Impingement Syndrome

General Comments:

The manuscript titled "The Role of Ultrasonography in Hip Impingement Syndrome" provides a concise review of the existing literature on the crucial role of ultrasonography in diagnosing femoroacetabular impingement (FAI) and determining the need for additional examinations. The paper discusses the importance of diligent imaging in diagnosing hip pain, particularly FAI, and highlights the limitations of traditional imaging modalities such as hip radiography and magnetic resonance arthrography. The manuscript also emphasizes the advantages of ultrasonography in providing dynamic assessment, guiding interventions, and facilitating hip injections. The overall structure and content of the manuscript are appropriate. However, there are a few areas that require clarification, revision, and further elaboration. The following comments are provided to improve the clarity, accuracy, and completeness of the manuscript.

Specific Comments:

1. Abstract:

The abstract provides a brief overview of the manuscript but lacks specific details about the findings and conclusions. It would be helpful to include a summary of the main findings and their implications in diagnosing and managing FAI using ultrasonography.

2. Introduction:

a. The introduction should provide a more comprehensive background on hip impingement syndrome, including its prevalence, clinical significance, and impact on patient outcomes. This would help readers better understand the relevance and importance of diagnosing FAI accurately.

b. The introduction briefly mentions hip radiography, magnetic resonance arthrography, and ultrasonography as imaging modalities but does not explain why ultrasonography is emerging as an alternative diagnostic tool. It would be beneficial to provide a succinct overview of the advantages and limitations of each imaging modality to justify the focus on ultrasonography.

3. Discussion of Imaging Modalities:

a. The manuscript briefly mentions that magnetic resonance arthrography is considered the gold standard examination but lacks a detailed explanation of its role in diagnosing FAI. It would be useful to elaborate on the specific features that make magnetic resonance arthrography valuable for diagnosing FAI and how it compares to other modalities.

b. The discussion of ultrasonography as an alternative diagnostic tool is adequate, but it would be beneficial to include specific examples of the ultrasonographic findings associated with FAI. Providing descriptions or images of the sonographic features of cam deformity, pincer deformity, and labral tears would enhance the understanding of the technique's diagnostic capabilities.

4. Ultrasonography Technique:

a. The manuscript mentions that ultrasonography of the hip joint is challenging due to its complex regional anatomy and deep location but does not explain how these challenges are overcome during the examination. It would be helpful to discuss the technical considerations, patient positioning, and probe selection necessary to obtain optimal images of the hip joint.

b. The manuscript briefly mentions that ultrasonography can provide detailed information about various hip quadrants but does not elaborate on the specific anatomical structures and pathologies that can be visualized in each quadrant. Providing a more detailed description of the sonographic evaluation of the anterior, medial, lateral, and posterior quadrants would enhance the readers' understanding of the technique's capabilities.

5. Ultrasonography-Guided Interventions:

a. The manuscript mentions that ultrasonography is beneficial for guided interventions around the hip joint but does not provide specific details about the types of interventions that can be performed under ultrasound guidance. It would be helpful to discuss the common therapeutic and diagnostic procedures, such as joint aspirations, injections, and biopsies, that can be facilitated by ultrasonography.

b. Additionally, it would be useful to discuss the advantages and potential limitations of ultrasonography-guided interventions compared to other guidance techniques, such as fluoroscopy etc.

nil

Author Response

We made changes to the introduction. Regarding discussing the benefits and limitations of the various imaging modalities of hip impingement syndromes. As instructed, we elaborated more and underlined the role of MR arthrography as the gold standard examination. Regarding section 4, the purpose of our work is to explore the use of US in hip impingement syndromes from an orthopaedic point of view, how often it is necessary and for which pathologies can be of help to be diagnosed. It is not our intention to focus on the technical US issues. That way, we did not discuss specifically technical US issues. Finally, regarding section 5, we discussed and compared ultrasound-guided injections with fluoroscopy-guided interventions, adding a new paragraph, and we discussed other hip procedures that could be facilitated with the employment of ultrasonography.

Round 2

Reviewer 1 Report

The reviewer considers that the manuscript has made a great improvement since the first revision. The methodological aspect, which was a big problem to even read the initial article, has been substantially improved. However, there are still important methodological issues that make it difficult to consider it for publication in an impact scientific journal. A REVIEW type article has been made, but there is no RESULTS section. This methodological question seems to me an essential requirement to later be able to review the article in depth and add the "small necessary" improvements. It does not make sense to review in depth a scientific text that does not have the minimum structure of any scientific article. For this reason, and despite the improvements introduced in this manuscript, I recommend following the guide for writing PRISMA reviews (https://systematicreviewsjournal.biomedcentral.com/articles/10.1186/s13643-021-01626-4).

Author Response

Regarding the reviewer's comments, we have written a narrative review type of manuscript and not a systematic review. That's why we have not followed the PRISMA guidelines, and we do not include a flowchart or a results section. I the reviewer thinks that we should remove the new materials section, we will do that. In any case, we aim for a review structure similar to the following manuscript, which was recently published by the diagnostics journal: Diagnostics | Free Full-Text | Early Optical Coherence Tomography Biomarkers for Selected Retinal Diseases—A Review (mdpi.com).

Reviewer 2 Report

Reviewer Report

I hope this letter finds you well. I have carefully reviewed the manuscript submitted by the authors. After thorough consideration of the revised version and comparing it with the original submission, I regret to inform you that the authors have not adequately addressed the concerns and queries I raised in my previous review. I would like to highlight the following major points of concern:

1. Lack of Point-to-Point Response: Despite my detailed queries and comments provided in the initial review, the authors have not responded adequately to the points raised. I do not see clear, specific responses to each query, which makes it challenging to assess if the issues have been appropriately addressed.

2. Inadequate Presentation of Changes: In the main manuscript, the authors have not clearly indicated where changes have been made, nor have they highlighted the modifications. It is essential for reviewers and readers to easily identify the alterations made to understand how the manuscript has been improved.

3. Absence of Supporting Information: The revised manuscript lacks additional explanations or information where the changes were implemented. Without proper guidance from the authors, it is difficult to verify the accuracy and validity of the modifications.

Given the above concerns, I am unable to recommend acceptance of the manuscript in its current form. To move forward, I kindly request the following actions from the authors:

a) Provide a comprehensive point-by-point response to the queries and comments raised by me in the previous review.

b) Clearly indicate and highlight the changes made throughout the main manuscript, allowing for easy identification and assessment of the revisions.

c) If any significant modifications have been made, provide additional information or explanations in the relevant sections of the manuscript to support the changes.

Once these issues have been adequately addressed and the necessary improvements have been made, I would be more than willing to re-evaluate the manuscript.

I remain available for any further questions or clarifications.

Sincerely,

nil

Author Response

During our revision, we uploaded a Word document with enabled track changes, but the way the revised manuscript was presented to the reviewers, it is obvious that they were not able to point out any changes. This time, we uploaded the same Word document, highlighting each main change, so I hope it will be more helpful. Regarding the point-to-point response, we tried to address most issues with our previous response letter. In the case that anything was missing, I'm presenting again our comments: We made changes to the introduction. Regarding discussing the benefits and limitations of the various imaging modalities of hip impingement syndromes. As instructed, we elaborated more and underlined the role of MR arthrography as the gold standard examination. Regarding section 4, the purpose of our work is to explore the use of US in hip impingement syndromes from an orthopaedic point of view, how often it is necessary and for which pathologies can be of help to be diagnosed. It is not our intention to focus on the technical US issues. That way, we did not discuss specifically technical US issues. Finally, regarding section 5, we discussed and compared ultrasound-guided injections with fluoroscopy-guided interventions, adding a new paragraph, and we discussed other hip procedures that could be facilitated with the employment of ultrasonography.)

Round 3

Reviewer 1 Report

The manuscript has improved considerably in its organization after suggestions made by other reviewers.

As I have commented in the previous answers, it seems to me that the content of the article is perfect, but that the main problem was methodological: as the authors comment, it is a narrative review, not a systematic review. Initially I understood that it was a systematic review, and in that case, the search criteria and results should have been much more developed, as well as the results. As the authors indicate in their answer, this is a narrative review, in my opinion it should be added in the TITLE ": a narrative review". In this case, the methodology section does not make sense to explain the search criteria, that two people analyzed the articles and a third resolved in case of doubt...

Here is an example: https://pubmed.ncbi.nlm.nih.gov/32471412/

For my part, it is not necessary to review anything else.

Author Response

Thank you very much for your feedback. The suggested changes have been updated.

Reviewer 2 Report

Based on the authors' thorough response to the reviewers' comments in the revised manuscript, I recommend accepting it for publication in its current form.

nil

Author Response

Thank you very much.